# Free vibration characteristics of trapezoidal nanoplate rested on viscoelastic substrate with arbitrary boundary conditions using differential quadrature method

**Ramin Abdellahi**[1], **Mohsen Esmaeili**[2], **Mirsami Yeganli**[3], **Ali Mokhtarian**[4], **Roohallah Alizadehsani**[5], **Paweł Pławiak**[6,7]*

**1** Department of Mechanical Engineering, Khomeinishahr Branch, Islamic Azad University, Khomeinishahr/Isfahan, Iran, **2** Department of Mechanical Engineering, Islamic Azad University, Tehran, Iran, **3** Department of Mechanical Engineering, Tehran University, Tehran, Iran, **4** Department of Mechanical Engineering, Khomeinishahr Branch, Islamic Azad University, Khomeinishahr/Isfahan, Iran, **5** Institute for Intelligent Systems Research and Innovation (IISRI), Deakin University, Waurn Ponds, Australia, **6** Department of Computer Science, Faculty of Computer Science and Telecommunications, Cracow University of Technology, Krakow, Poland, **7** Institute of Theoretical and Applied Informatics, Polish Academy of Sciences, Gli-wice, Poland

* pawel.plawiak@pk.edu.pl

## Abstract

This study investigates the free vibration analysis of trapezoidal nanoplate resting on viscoelastic foundation based on first order shear deformation theory (FSDT) incorporating nonlocal elasticity theory, using differential quadrature (DQ) method. The nanoplate's governing equations of motion together with various associated boundary conditions have been discretized applying a mapping DQ method in the spatial domain. Then the complex natural frequencies of the trapezoidal nanoplates obtained by solving the eigen value matrix equation. Verification of the study is confirmed by comparing its numerical results with those available in the literature, then parametric study is thoroughly performed. A special attention is drawn to the role of geometrical parameters of nanoplate, stiffness and damping parameters of foundation, nonlocal parameter and boundary condition on natural frequencies characteristics. This research's results are useful for designing of the nano-electromechanical systems (NEMS) efficiently and show the potential application of the system as highly sensitive nano-sensors and resonator in damped medium.

## 1 Introduction

In recent years, nanotechnology has found a substantial role in our life due to the wide scope of its potential application in many nanostructures like NEMS. Graphene nanoplates are a class of nanostructures with 2D shape with superior mechanical properties and great potential for extensive application in NEMS. Considering their dynamic and vibrational characteristics like frequency response, resonance phenomena and instabilities are essential in engineering applications in order to make optimal usage of these systems for designing and developing NEMS such as nano-actuators, nano-harvesters, nano-composites and nano-resonators [1].

**Data availability statement:** All relevant data for this study are within the paper.

**Funding:** The author(s) received no specific funding for this work.

**Competing interests:** The authors have declared that no competing interests exist.

It is possible to combine graphene sheets into complex nano-structures and apply them in NEMS mainly as sensors [2] and high frequency resonators [3] considering vibration damping in the system.

In the field of mechanical and vibrational behaviour of rectangular nanoplate, The finite strip method was considered by Sarrami-Foroushani and Azhari [4] to study buckling load of single and multi-layered rectangular graphene sheets (GS) counting interlayer effects at various boundary conditions. Liu and Chen [5] reported dynamic analysis of the periodic nanoplates applying the FSDT and nonlocal theory at different boundary conditions. Radebe and Adali [6] investigated the size dependent buckling and free vibration of orthotropic rectangular nanoplates using the nonlocal theory, with considering the uncertain mechanical properties. Based on the higher order shear deformation theory (HSDT) and nonlocal theory, Mehar et al. [7] studied free vibration of nanoplates using FEM. Recently, Nguyen Thi et al. [8] (2024) studied the mechanical response of nanoplates by accounting both flexoelectric and nonlocal stress effects. The natural frequency of fluid-infiltrated porous nanoplates under different boundary conditions was determined using the Galerkin-Vlasov method based on improved FSDT.

Since in the real-world application, the nanostructures must be considered in medium with damping features, The influence of viscoelastic substrate has been considered on the vibration characteristics of nano-systems in the literature. The vibration characteristics of simply-supported nanoplate resting on viscoelastic foundation was studied using the nonlocal plate theory by Pouresmaeeli et al. [9] analytically. According to CPT and nonlocal elasticity theory, Hosseini Hashemi et al. [10] obtained the forced vibration responses of rectangular nanoplates resting on viscoelastic medium using Kelvin Voight model, applying the analytical method. Khanmirza et al. [11] considered vibration of magneto electro-elastic nanoplate mass sensor resting on visco-Pasternak substrate. Considering hygrothermal effects, Shahsavari et al. [12] examined a dynamic analysis response for nanoplate resting on visco-Pasternak substance and subjected to moving mass based on nonlocal theory and CPT. Hosseini et al. [13] obtained buckling loads and natural frequencies of FG nanoplates mounted on visco-elastic substrate, considering the nonlocal and surface effects, respectively. Moradi et al. [14] investigated the vibration characteristics of FG annular and circular nanoplates while rested on a Viscoelastic foundation utilizing the DQ technique. Ebrahimi and Hosseini [15] investigated nonlinear dynamic stability of nanoplates while rested on a viscoelastic substrate, incorporating surface stress theories with nonlocal elasticity relations. A finite element formulation based on nonlocal theory have been applied by Pham et al. [16] to study the vibration characteristics of visco-elastic orthotropic nanoplates resting on variable visco-elastic foundations based on refined HSDT. Applying the Laplace transform method, vibrational and dynamic characteristics of thermoelastic nanoplate resting on viscoelastic foundation have been studied by Zhao et al. [17] based on the fractional order of viscoelastic model, combining the nonlocal effects and strain gradient elasticity. Recently at 2024, Thom et al. [18] applied analytical methods to examine the static bending response, thermal buckling, vibration behaviour, and forced vibration characteristics of nanoplates resting on viscoelastic foundation, considering the flexomagnetic effect.

Arbitrary shaped nanoplates such as skew and trapezoidal ones are generally employed in NEMS devices like actuators and sensors. Hence, to correctly design and manufacture nano-devices, a thorough knowledge of the mechanical and vibrational behaviors of these oblique nano-structured materials is essential. The larger part of the research papers has been done on the mechanical and vibration behavior of rectangular shaped sheets, and limited studies has been done on arbitrary quadrilateral shapes in the literature. For example in macrostructures with classical continuum theory; Karami et al. [19] applied DQ method to study

both static and dynamic characteristics of skew and trapezoidal composite plates. DQ technique have been applied to obtain the natural frequencies of trapezoidal plates using FSDT by Zamani et.al [20]. Malekzadeh and Zarei [21] examined the vibrational behaviour of arbitrary four-sided composite plates. Torabi et al. [22] established the unified formulation based on the HSDT and VDQ method along with coordinate transformation technique to study the linear thermal buckling of various shapes of composite plates (like skew, quadrilateral, triangular and circular) reinforced with functionally graded CNTs.

In connection with vibration analysis of trapezoidal and skewed nanoplates; Free vibration analysis of arbitrary quadrilateral shaped nanoplates were studied by Malekzadeh et al. [23] based on the nonlocal elasticity theory and FSDT, using the DQ method. Using the same theories, Alibeygi Beni and Malekzadeh [24] analyzed the natural frequencies of skew nanoplates, considering various boundary conditions. Applying the Galerkin method, Babaie and Shahidi studied the size-dependent natural frequencies of quadrilateral SLGS [25] to examine the nano-scale effect. Combining the nonlocal theory and surface effects, Malekzadeh et al. [26] studied nonlinear vibrations of skew nanoplate using the CPT. Employing the kp-Ritz element free method, Hang et al. [27] investigated the vibration of quadrilateral SLGS with different boundary conditions, in magnetic field using CPT and nonlocal elasticity. Considering the Gurtin–Murdoch surface theory and three-dimensional (3D) elasticity relations, natural frequency analysis of FG quadrilateral nanoplates in thermal environment studied by means of the variational DQ method by Shahabodini et al. [28]. Applying Nonlocal theory and Kirchhoff model, thermomechanical vibration features and buckling loads of quadrilateral smart piezoelectric nanoplates, have been considered at various boundary conditions with spline finite strip method by Analooei et al. [29]. Applying the Ritz method, Nonlinear vibration behavior of the trapezoidal rotating FG microplates in thermal environment were studied using MSGT considering four-variable refined plate theory by Shenas et al. [30] (2022).

Analysis of other mechanical properties of trapezoidal and skewed nanoplates have been reported in the literature; For example, using Eringen's nonlocal thory, Yuan et al. [31] (2020) examined the critical shear buckling load of FG skew nanoplates with the aid of diverse forms of the homogenization scheme. Shear buckling characteristics of FGM skew nano plates was obtained considering surface stress effect and HSDT, by means of the Ritz method with Gram-Schmidt shape functions [32]. Utilizing DQ method, dynamic stability of quadrilateral viscoelastic SLGS with movable boundaries and having various types of defects, like vacancy defects, has been evaluated using Strain Gradient Theory and HSDT. Also, the effects of temperature change and external magnetic field on the stability behavior have been studied [33]. Newly, Wang and Liu [34] studied critical buckling loads of skew plates with elastically controlled edges using the Ritz method, where the shape function is expressed as Legendre polynomials.

In the recent years, numerous studies have been reported in the fields of size-dependent vibrational characteristics of skew and trapezoidal shaped nanostructures. In the large number of available works, the effect of viscoelastic medium has been ignored, while in a real application the structure must be considered in medium with damping features like viscoelastic foundation. In connection with the subject of size-dependent vibration of trapezoidal nanoplates, to the best of authors' knowledge, no papers have been reported in the literature concerning the influence of viscoelastic substrate on vibrational behavior of the trapezoidal nanoplate resting on it. In the present study, the size dependent natural frequencies of trapezoidal nanoplate resting on viscoelastic foundation with different combinations of B.C.s have been conducted by means of FSDT and DQ method. the structure proposed as NEMS system with damping effects. The main contribution of the present study is considering the influences of viscoelastic foundation in the examination of free vibration behaviour of trapezoidal-shaped

nanoplates, having high shape diversity, with different combinations of Boundary conditions (B.Cs).

The present study is organized into four primary sections. Section 2 illustrated the mathematical modeling of nanoplate, the constructive relations and its governing equations of motion. These governing equations have been transformed from the trapezoidal physical domain into the rectangular computational domain in the Section 3 by means of mathematical operations and then rewritten in the form of generalized eigenvalue problem by means of DQ method for various B.C. Finally at section 4, the validation of numerical results presented and parametric study have been shown by means of plots.

## 2 Mathematical modeling

The schematic illustration of an embedded trapezoidal nano-plate is shown in Fig 1a. As presented in Fig 1b, a trapezoidal nanoplate with specific geometrical parameters resting on viscoelastic foundation has been shown. The $x$, $y$ and $z$ coordinates of the axes are taken along the two in plane axis and thickness of the nano-plate, respectively. The displacement field of plate has been considered based on the FSDT as follow:

$$u = u_0 + z\,\varphi_x;\; v = v_0 + z\,\varphi_y;\; w = w_0 \qquad (1)$$

Here, $u_0$, $v_0$ and $w_0$ represent the mid-plane displacement along $x$, $y$ and $z$ directions; also $\varphi_x$ and $\varphi_y$ are rotations of the mid-plane's normal about $y$ and $x$ axes, respectively. The 2D strains based on Eq. (1) are as follows:

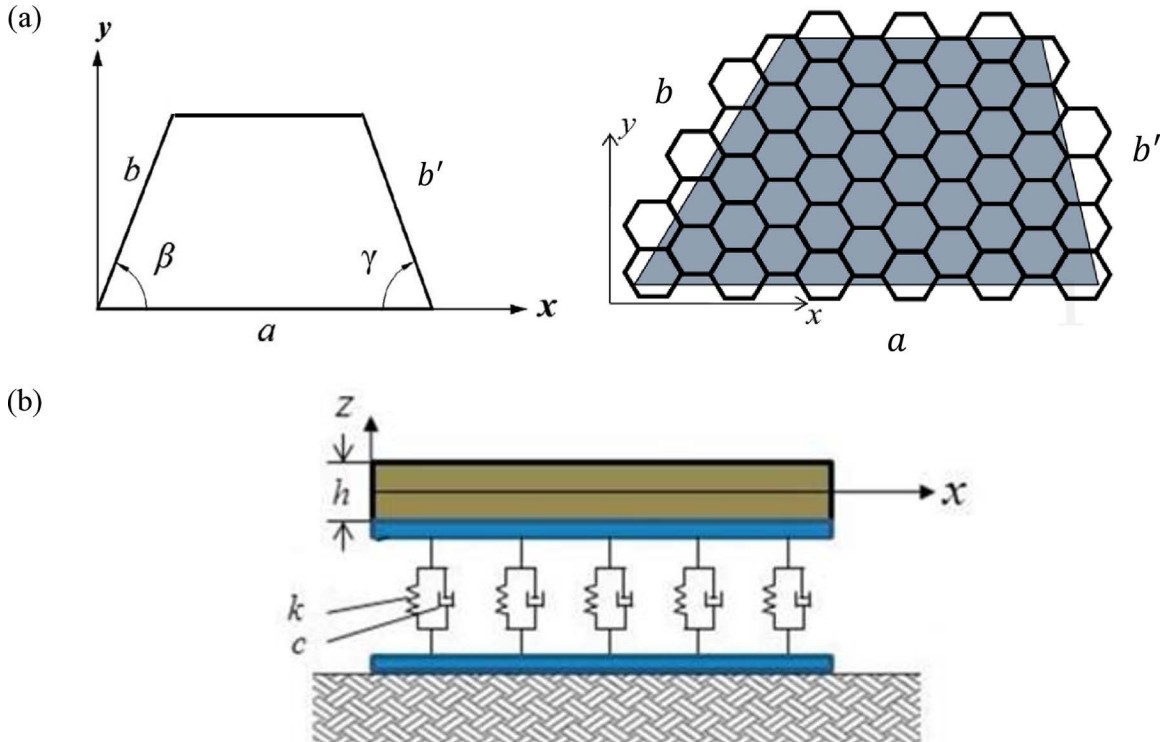

**Fig 1. Schematic view of:** (a) continuum and geometric model of GS as trapezoidal nanoplate, (b) quadrilateral nanoplate resting on viscoelastic substrate.

$$\varepsilon_{xx} = \frac{\partial u_0}{\partial x} + z\frac{\partial \varphi_x}{\partial x}; \; \varepsilon_{yy} = \frac{\partial v_0}{\partial y} + z\frac{\partial \varphi_y}{\partial x};$$

$$\varepsilon_{xy} = \frac{1}{2}\left(\frac{\partial u_0}{\partial y} + \frac{\partial v_0}{\partial x} + z\frac{\partial \varphi_x}{\partial y} + z\frac{\partial \varphi_y}{\partial x}\right)$$

(2)

Based on the nonlocal theory, the stress-strain relations are:

$$\left(1 - \mu\nabla^2\right)\sigma_{ij}^{(nl)} = \sigma_{ij}^{(l)} = S_{ijkl}\varepsilon_{kl} \tag{3}$$

In which, $\sigma_{ij}^{(nl)}$, $\sigma_{ij}^{(l)}$ are the tensors of nonlocal and local stresses, respectively; $\varepsilon_{ij}$ indicates the strain tensor and $S_{ijkl}$ is the tensor of elasticity. Using Eqs. (1-3), the nonlocal stress and moment resultants are given as ( $M_{ij}$ and $Q_i$ are resultant moment and shear force vectors in the cartesian coordinates):

$$\left(1 - \mu\nabla^2\right)\left\{M_{xx} \quad M_{yy} \quad M_{xy}\right\}^T = \begin{bmatrix} D_{11} & D_{12} & 0 \\ D_{12} & D_{22} & 0 \\ 0 & 0 & D_{66} \end{bmatrix}\left\{\begin{bmatrix} \frac{\partial \varphi_x}{\partial x} & \frac{\partial \varphi_y}{\partial y} & \left(\frac{\partial \varphi_y}{\partial x} + \frac{\partial \varphi_x}{\partial y}\right) \end{bmatrix}\right\}^T \quad \text{(4a)}$$

$$\left(1 - \mu\nabla^2\right)\left\{Q_x \quad Q_y\right\}^T = \begin{bmatrix} A_{55} & 0 \\ 0 & A_{44} \end{bmatrix}\left\{\begin{bmatrix} \left(\varphi_x + \frac{\partial w}{\partial x}\right) & \left(\varphi_y + \frac{\partial w}{\partial y}\right) \end{bmatrix}\right\}^T \quad \text{(4b)}$$

In which the coefficients are equals to: $\left(A_{ij}, D_{ij}\right) = \int_{h/2}^{-h/2}\left(1, z, z^2\right)C_{ij}dz$ . The $C_{ij}$ coefficients are defined as bellow:

$$C_{11} = \frac{E_{11}}{1 - \nu_{12}\nu_{21}} \; ; \; C_{12} = C_{21} = \frac{\nu_{12}E_{11}}{1 - \nu_{12}\nu_{21}} \; ; \; C_{22} = \frac{E_{22}}{1 - \nu_{12}\nu_{21}}; C_{66} = \frac{E_{11}}{2\left(1 + \nu_{12}\right)} \tag{5}$$

$E_{11}$ and $E_{22}$ are elastic modules and $\nu_{12}$ and $\nu_{21}$ are poison ratios of orthotropic nanoplate. The governing equations of motion for can be obtained by means of the Hamilton's principle as:

$$\frac{\partial N_{xx}}{\partial x} + \frac{\partial N_{xy}}{\partial y} = I_0\frac{\partial^2 u}{\partial t^2}; \tag{6a}$$

$$\frac{\partial N_{xy}}{\partial x} + \frac{\partial N_{yy}}{\partial y} = I_0\frac{\partial^2 v}{\partial t^2}; \tag{6b}$$

$$\frac{\partial M_{xx}}{\partial x} + \frac{\partial M_{xy}}{\partial y} - Q_x = I_2\frac{\partial^2 \varphi_x}{\partial t^2} \tag{6c}$$

$$\frac{\partial M_{xy}}{\partial x} + \frac{\partial M_{yy}}{\partial y} - Q_y = I_2\frac{\partial^2 \varphi_y}{\partial t^2} \tag{6d}$$

$$\frac{\partial Q_x}{\partial x} + \frac{\partial Q_y}{\partial y} + f + N_{xx}\frac{\partial^2 w}{\partial x^2} + 2N_{xy}\frac{\partial^2 w}{\partial x\partial y} + N_{yy}\frac{\partial^2 w}{\partial y^2} = I_0\frac{\partial^2 w}{\partial t^2} \tag{6e}$$

Considering Eqs. (6c) - (6e), (4a) - (4b) and ignoring the in-plane forces ($N_{xx} = N_{xy} = N_{yy} = 0$), the next equations for vibrating nano-plate will be obtained:

$$D_{11}\frac{\partial^2 \varphi_x}{\partial x^2} + D_{12}\frac{\partial^2 \varphi_y}{\partial y\partial x} + D_{66}\left(\frac{\partial^2 \varphi_x}{\partial y^2} + \frac{\partial^2 \varphi_y}{\partial y\partial x}\right) - k_sA_{55}\left(\varphi_w + \frac{\partial w}{\partial x}\right)$$

$$-k_sA_{45}\left(\varphi_y + \frac{\partial w}{\partial y}\right) = I_2\frac{\partial^2}{\partial t^2}\left[\varphi_x - \mu\left(\frac{\partial^2 \varphi_x}{\partial x^2} + \frac{\partial^2 \varphi_x}{\partial y^2}\right)\right] \tag{7a}$$

$$D_{22}\frac{\partial^2 \varphi_y}{\partial y^2} + D_{12}\frac{\partial^2 \varphi_x}{\partial y \partial x} + D_{66}\left(\frac{\partial^2 \varphi_y}{\partial x^2} + \frac{\partial^2 \varphi_x}{\partial y \partial x}\right) - k_s A_{44}\left(\varphi_y + \frac{\partial w}{\partial y}\right)$$

$$-k_s A_{45}\left(\varphi_x + \frac{\partial w}{\partial x}\right) = I_2 \frac{\partial^2}{\partial t^2}\left[\varphi_y - \mu\left(\frac{\partial^2 \varphi_y}{\partial x^2} + \frac{\partial^2 \varphi_y}{\partial y^2}\right)\right] \quad (7b)$$

$$k_s A_{55}\left(\frac{\partial \varphi_x}{\partial x} + \frac{\partial^2 w}{\partial x^2}\right) + k_s A_{44}\left(\frac{\partial \varphi_y}{\partial y} + \frac{\partial^2 w}{\partial y^2}\right) + k_s A_{45}\left(\frac{\partial \varphi_x}{\partial y} + \frac{\partial \varphi_y}{\partial x} + 2\frac{\partial^2 w}{\partial y \partial x}\right)$$

$$-f + \mu\left(\frac{\partial^2 f}{\partial x^2} + \frac{\partial^2 f}{\partial y^2}\right) = I_0 \frac{\partial^2}{\partial t^2}\left[w - \mu\left(\frac{\partial^2 w}{\partial x^2} + \frac{\partial^2 w}{\partial y^2}\right)\right] \quad (7c)$$

In which the force acting on nanoplate from viscoelastic substrate are $f = kw + c\frac{\partial w}{\partial t}$, where $k$ and $c$ are stiffness and damping coefficients of viscoelastic foundation, respectively; and $k_s = 5/6$

Different boundary conditions (B.Cs) for edges are as [35]:

$$\text{Free}(F): M_{nn} = Q_n = M_{ns} = 0 \quad (8a)$$

$$\text{Simply supported}(S): M_{nn} = \varphi_s = w = 0 \quad (8b)$$

$$\text{Clamped}(C): \varphi_n = \varphi_s = w = 0 \quad (8c)$$

Where normal and tangential directions have been represented by n and s, respectively. Besides $M_{nn}$ $M_{ns}$ and $Q_n$ are bending moment, twisting moment, and shear force acting on the boundary in the z direction, respectively. The mentioned parameters in the cartesian coordinate equals to:

$$M_{nn} = M_{xx}n_x^2 + M_{yy}n_y^2 + 2M_{xy}n_x n_y \quad (9a)$$

$$M_{ns} = n_x n_y\left(M_{yy} - M_{xx}\right) + M_{xy}\left(n_x^2 - n_y^2\right) \quad (9b)$$

$$Q_n = n_x Q_x + n_y Q_y \quad (9c)$$

Where $n_x$ and $n_y$ are the x and y components of the vector normal to the edge, respectively.

## 3 Mapping and solution procedure

### 3.1 Geometric mapping

The relation between the x-y coordinate system (physical domain) and the mapped $\zeta - \eta$ coordinate system (computational domain), are as follows:

$$x = \zeta + \eta\cos(\gamma) - \frac{\eta \zeta \sin(\beta - \gamma)}{a\sin(\beta)}; \; y = \eta\sin(\gamma) \quad (10)$$

Based on the mathematical relations for first and second-order derivatives of any function, in transformation of coordinate system and transformation Jacobian matrix (As represented in the Appendix A1, A2), the governing equations have been transformed from the physical domain into the computational domain. The transformed of nanoplate's Eqs. (7a)-(7c) are as follows:

$$b_1\frac{\partial \varphi_x}{\partial \zeta} + b_2\frac{\partial \varphi_x}{\partial \eta} + s_{11}\frac{\partial^2 w}{\partial \zeta^2} + b_3\frac{\partial \varphi_y}{\partial \zeta} + b_4\frac{\partial \varphi_y}{\partial \eta} + s_{21}\frac{\partial^2 w}{\partial \zeta^2} + s_{12}\frac{\partial^2 w}{\partial \eta^2} + s_{13}\frac{\partial^2 w}{\partial \zeta \partial \eta}$$

$$-a_{11}\frac{\partial w}{\partial \zeta} - a_{12}\frac{\partial w}{\partial \eta} + s_{22}\frac{\partial^2 w}{\partial \eta^2} + s_{23}\frac{\partial^2 w}{\partial \zeta \partial \eta} - a_{21}\frac{\partial w}{\partial \zeta} - a_{22}\frac{\partial w}{\partial \eta} \quad (11a)$$

$$-\frac{1}{k_s A_{44}}\left(k + c\frac{\partial}{\partial t} + I_0\frac{\partial^2}{\partial t^2}\right)\Gamma(w) = 0$$

$$D_{11}\left(s_{11}\frac{\partial^2\varphi_x}{\partial\zeta^2}+s_{12}\frac{\partial^2\varphi_x}{\partial\eta^2}+s_{13}\frac{\partial^2\varphi_x}{\partial\zeta\partial\eta}-a_{11}\frac{\partial\varphi_x}{\partial\zeta}-a_{12}\frac{\partial\varphi_x}{\partial\eta}\right)-k_sA\left(\varphi_x+b_1\frac{\partial w}{\partial\zeta}+b_2\frac{\partial w}{\partial\eta}\right)$$

$$+\left(D_{12}+D_{66}\right)\left(s_{31}\frac{\partial^2\varphi_y}{\partial\zeta^2}+s_{32}\frac{\partial^2\varphi_y}{\partial\eta^2}+s_{33}\frac{\partial^2\varphi_y}{\partial\zeta\partial\eta}-a_{31}\frac{\partial\varphi_y}{\partial\zeta}-a_{32}\frac{\partial\varphi_y}{\partial\eta}\right) \qquad (11b)$$

$$+D_{66}\left(s_{21}\frac{\partial^2\varphi_x}{\partial\zeta^2}+s_{22}\frac{\partial^2\varphi_x}{\partial\eta^2}+s_{23}\frac{\partial^2\varphi_x}{\partial\zeta\partial\eta}-a_{21}\frac{\partial\varphi_x}{\partial\zeta}-a_{22}\frac{\partial\varphi_x}{\partial\eta}\right)=I_2\frac{\partial^2\Gamma\left(\varphi_x\right)}{\partial t^2}$$

$$D_{22}\left(s_{21}\frac{\partial^2\varphi_y}{\partial\zeta^2}+s_{22}\frac{\partial^2\varphi_y}{\partial\eta^2}+s_{23}\frac{\partial^2\varphi_y}{\partial\zeta\partial\eta}-a_{21}\frac{\partial\varphi_y}{\partial\zeta}-a_{22}\frac{\partial\varphi_y}{\partial\eta}\right)-k_sA\left(\varphi_y+b_3\frac{\partial w}{\partial\zeta}+b_4\frac{\partial w}{\partial\eta}\right)$$

$$+\left(D_{12}+D_{66}\right)\left(s_{31}\frac{\partial^2\varphi_x}{\partial\zeta^2}+s_{32}\frac{\partial^2\varphi_x}{\partial\eta^2}+s_{33}\frac{\partial^2\varphi_x}{\partial\zeta\partial\eta}-a_{31}\frac{\partial\varphi_x}{\partial\zeta}-a_{32}\frac{\partial\varphi_x}{\partial\eta}\right)$$

$$+D_{66}\left(s_{11}\frac{\partial^2\varphi_y}{\partial\zeta^2}+s_{12}\frac{\partial^2\varphi_y}{\partial\eta^2}+s_{13}\frac{\partial^2\varphi_y}{\partial\zeta\partial\eta}-a_{11}\frac{\partial\varphi_y}{\partial\zeta}-a_{12}\frac{\partial\varphi_y}{\partial\eta}\right)$$

$$=I_2\frac{\partial^2\Gamma\left(\varphi_y\right)}{\partial t^2}. \qquad (11c)$$

which:

$$\begin{bmatrix}b_1 & b_2\\b_3 & b_4\end{bmatrix}=\begin{bmatrix}[j]^{-1}(1,1) & [j]^{-1}(1,2)\\ [j]^{-1}(2,1) & [j]^{-1}(2,2)\end{bmatrix} \qquad (12a)$$

$$s_{mn}=\left[j^{(2)}\right]^{-1}(m,n);\ a_{mn}=\left[j^{(2)}\right]^{-1}\left[j^{(1)}\right][j]^{-1}(m,n) \qquad (12b)$$

Also, the operator $\Gamma(\blacksquare)$ reperesented in the appendix (A3).

## 3.2 Numerical solution by DQ method

In this section, the main steps toward numerical solution based on DQM are expressed. Based on the DQM, the m-th derivative of the function $V(\zeta,\eta)$ would be acknowledged as [19]:

$$\frac{\partial^m V}{\partial x^m}\bigg|_{\zeta=\zeta_i,\eta=\eta_j}=\sum_{p=1}^{N_\zeta}\sum_{k=1}^{N_\eta}g_{ip}^{(m)}I_{pk}^\zeta V_{kj};\ \left(1\le i\le N_\zeta\right)$$

$$\frac{\partial^m V}{\partial y^m}\bigg|_{\zeta=\zeta_i,\eta=\eta_j}=\sum_{p=1}^{N_\zeta}\sum_{k=1}^{N_\eta}I_{ip}^\eta h_{pk}^{(m)}V_{kj};\ \left(1\le i\le N_\eta\right) \qquad (13)$$

In which, $N_\zeta$ and $N_\eta$ denote total number of discrete points considered along $\zeta$ and $\eta$ axes. Moreover $g_{ip}^{(m)}$, $h_{pk}^{(m)}$, are weighting coefficients of the $m$-th partial derivative at point (i, j), represented in the Ref. [19]. Also, identity matrix is shown by $I_{pk}^\zeta$ and $I_{pk}^\eta$. In this paper, the non-uniform Chebyshev grid points distribution is assumed, for which the coordinates of grid points $(\zeta_i,\eta_j)$ along the reference surface are:

$$\xi_i=\frac{L_\zeta}{2}\left(1-\cos\left(\frac{i-1}{N_\zeta-1}\pi\right)\right);\ \eta_j=\frac{L_\eta}{2}\left(1-\cos\left(\frac{i-1}{N_\eta-1}\pi\right)\right) \qquad (14)$$

Where $L_\zeta$, $L_\eta$ are the transformed length of plate in the $\zeta$ and $\eta$ directions, respectively. Applying GDQ method (Eqs. (13)) to Eqs. (11a)-(11c), one can write the subsequent set of algebraic equations as:

$$[K]\{q\}=[C]\{\partial q/\partial t\}+[M]\{\partial^2 q/\partial t^2\} \qquad (15)$$

In which [K] denotes the stiffness matrix; [C] is the damping matrix; [M] is the mass matrix; and {q} is vector of DOFs including values of w, $\varphi_\zeta$ and $\varphi_\zeta$ at all nodes.

For finding the natural frequencies, the nodes in all domain and the boundaries have been separated. Equations of motion (Eq.15) and boundaries equations have been written in the matrix form as follow [36]:

$$\left\{\begin{bmatrix} [K_{bb}] & [K_{bd}] \\ [K_{db}] & [K_{dd}] \end{bmatrix}\right\}\begin{Bmatrix} q_b \\ q_d \end{Bmatrix} + \left\{\begin{bmatrix} [C_{bb}] & [C_{bd}] \\ [C_{db}] & [C_{dd}] \end{bmatrix}\right\}\begin{Bmatrix} \partial q_b/\partial t \\ \partial q_b/\partial t \end{Bmatrix} + \left\{\begin{bmatrix} [M_{bb}] & [M_{bd}] \\ [M_{db}] & [M_{dd}] \end{bmatrix}\right\}\begin{Bmatrix} \partial^2 q_b/\partial t^2 \\ \partial^2 q_d/\partial t^2 \end{Bmatrix} = 0 \qquad (16)$$

Where subscript b and d show the boundary and domain points, respectively. Removing the points associated by the boundary in Eq. (16), then equating the mass and damping matrices to zero in boundary points, one can write $q_b = -\left([K_{bb}]^{-1}[K_{bd}]\right)q_d$ Then Eq. (16) can be rewritten as:

$$[K^*]\{Q_d\} + [C^*]\{\dot{Q}_d\} + [M^*]\{\ddot{Q}_d\} = 0 \qquad (17)$$

In which:

$$[K^*] = K_{dd} - K_{db}K_{bb}^{-1}K_{bd} \qquad (18a)$$

$$[C^*] = C_{dd} - C_{db}C_{bb}^{-1}C_{bd} \qquad (18b)$$

$$[M^*] = M_{dd} - M_{db}M_{bb}^{-1}M_{bd} \qquad (18c)$$

The system's DOFs have been presumed as $\{q\} = \{Q(\zeta,n)\}e^{i\omega t}$ where ω is the natural frequency. Rearranging the equations into the generalized eigenvalue problem yield:

$$\begin{bmatrix} [0] & [I] \\ -\left[(M^*)^{-1}[K^*]\right] & -\left[(M^*)^{-1}[C^*]\right] \end{bmatrix}\begin{Bmatrix} Q_d \\ Q_b' \end{Bmatrix} = \omega\begin{Bmatrix} Q_d \\ Q_b' \end{Bmatrix} \qquad (19)$$

Finally, Eq. (19) have been solved by means of the iterative method to obtain the eigenvalues as natural frequencies of the system.

## 4  Results and discussions

In this section, numerical results of the free vibration analyses of trapezoidal nanoplate resting on viscoelastic substrate with different B.Cs, geometrical parameters, foundation parameters, nonlocal parameters and mode number are presented through some examples. Additionally, Mechanical properties of Orthotropic nanoplate are as follows: $E_{11} = 1765\,\text{GPa}$, $E_{22} = 1588\,\text{GPa}$, $\nu_{12} = 0.3$ and $\rho = 2300\ \text{kg/m}^3$. Furthermore, the nondimensional parameters Ω, K and C are: $\Omega = \omega\sqrt{D_{11}/\rho ha^4}$, $K = ka^4/D_{11}$, $C = ca^2/\sqrt{\rho h D_{11}}$

### 4.1  Convergence study and verification

Since the number of assumed grid points can affect the results, the convergence of the results with respect to number of grid points (N) is studied. In Table 1 convergence of the dimensionless natural frequencies $\left(\Omega = \omega a^2\sqrt{\rho h/D_{11}}/\pi^2\right)$ by applying the DQ method have been studied for the trapezoidal nanoplate with: a/b = 4, β = 120, γ=60 with K = 100, C = 5. Results are set for different boundary conditions. The eigenvalue problem solved by different mesh sizes. Based on the numerical results, the DQ method obtained the accurate results even using a few grid points. Also, by increasing the number of grid points, the results converged to the final values, rapidly. Therefore, the mesh size of (14 × 14) have been used in the following results.

The first three dimensionless natural frequency of a trapezoidal plate with geometrical parameters: $b/a = 0.8$, $b'/a = 0.7$, $\beta = 70$, $\gamma = 75$ are evaluated and compared with results from

Ref [21], where used local theory, $\mu=0$) for two different values of h/a ratio (both thin and thick plates) at Table 2. By increasing the h/a ratio, the frequency parameter of all modes decreased.

Table 3 demonstrates the first three dimensionless natural frequencies $\left(\Omega = \omega a^2 \sqrt{\rho/E_2}/h\right)$ of a [30 60] laminated trapezoidal plate ($\mu=0$) with a = 1,b = 0.5 and β=90 at two corner angles $(\gamma)$ for various B.Cs. Material properties are: $(E_{11},G)/E_2 = (40,0.6), \nu_{12} = 0.25, \rho = 2500Kg/m^3$ The results have good agreements with reported results in Ref [20]. Besides numerical results showed by enhancing the corner angle ($\gamma$), natural frequency have been reduced. Besides, the frequency value drops significantly in the plates with replacing clamped boundary with free boundary.

Table 4 presented the first five natural frequencies of skew plates ($\mu=0$) with b/a = 1, h/b = 0.2 and β=45 at two different boundary conditions, CFCF (bridge-type) and CFFF (cantilever-type). The results also have good agreements with reported results in Ref. [20]. Based on the results, the fundamental natural frequencies of CFCF have been reduced about 84% by removing one clamped BC. Consequently, type of BCs is very crucial in the free vibration behaviour.

Table 5 represents the dimensionless natural frequencies of SSSS and CCCC orthotropic skew nanoplates for first two vibration mode with $E_1 = 10 E_2$, $G_{12} = 0.6 E_2$ and h/a = 0.05 at

**Table 1.** (a): Effect of B.Cs and number of points on Convergence study for first natural frequency. (b): Effect of B.Cs and number of grid points on Convergence study for second frequency.

| B.Cs | Number of Grid points | | | | | | |
|------|------|------|------|------|------|------|------|
| | 6 | 8 | 10 | 12 | 14 | 16 | 18 |
| (a) | | | | | | | |
| CCCC | 1.8894 | 1.8067 | 1.8059 | 1.8058 | 1.8058 | 1.8058 | 1.8058 |
| CSCS | 1.4965 | 1.4654 | 1.4637 | 1.4635 | 1.4635 | 1.4635 | 1.4634 |
| SSSS | 1.3068 | 1.3091 | 1.3079 | 1.3077 | 1.3077 | 1.3076 | 1.3076 |
| CCCF | 1.2935 | 1.2565 | 1.2546 | 1.2532 | 1.2523 | 1.2519 | 1.2516 |
| SSSF | 1.0455 | 1.0355 | 1.0335 | 1.0334 | 1.0333 | 1.0332 | 1.0332 |
| CFCF | 1.272 | 1.228 | 1.2276 | 1.2274 | 1.2272 | 1.227 | 1.227 |
| SFSF | 1.0185 | 1.0216 | 1.0218 | 1.0221 | 1.0221 | 1.0222 | 1.0222 |
| CCFF | 0.9987 | 0.9962 | 0.9868 | 0.9888 | 0.9899 | 0.9907 | 0.9905 |
| (b) | | | | | | | |
| CCCC | 2.3315 | 2.1923 | 2.18392 | 2.1848 | 2.1847 | 2.1847 | 2.1847 |
| CSCS | 2.0596 | 1.9761 | 1.9718 | 1.9724 | 1.9723 | 1.9723 | 1.9723 |
| SSSS | 1.6602 | 1.6794 | 1.67713 | 1.6769 | 1.6767 | 1.6767 | 1.6767 |
| CCCF | 1.7889 | 1.7294 | 1.7182 | 1.7163 | 1.7143 | 1.7132 | 1.7126 |
| SSSF | 1.3359 | 1.3398 | 1.3341 | 1.3325 | 1.3315 | 1.3309 | 1.3306 |
| CFCF | 1.659 | 1.5989 | 1.58977 | 1.5887 | 1.5882 | 1.5881 | 1.588 |
| SFSF | 1.2804 | 1.3294 | 1.32849 | 1.3276 | 1.327 | 1.3265 | 1.3262 |
| CCFF | 1.1155 | 1.1612 | 1.13546 | 1.1255 | 1.1102 | 1.1335 | 1.1205 |

**Table 2.** With Ref [21].

| Method | h/a = 0.001. | | | h/a = 0.2 | | |
|--------|------|------|------|------|------|------|
| | 1st | 2nd | 3rd | 1st | 2nd | 3rd |
| FEM [21] | 6.8310 | 13.198 | 14.374 | 4.2880 | 6.8910 | 7.3110 |
| Ritz [21] | 6.8270 | 13.131 | 14.340 | 4.2870 | 6.8880 | 7.3070 |
| DQM (present) | 6.8270 | 13.131 | 14.340 | 4.2873 | 6.8876 | 7.3066 |

**Table 3. Validation of first three frequencies of trapezoidal plate at various B.C and side angles.**

| γ | B.Cs | Frequency | | | | | |
|---|---|---|---|---|---|---|---|
| | | 1st mode | | 2nd mode | | 3rd mode | |
| | | Present | Ref [20] | Present | Ref [20] | Present | Ref [20] |
| 45 | CCCC | 16.865 | 16.866 | 23.615 | 23.618 | 30.164 | 30.199 |
| | SSSS | 12.067 | 11.716 | 19.250 | 19.154 | 26.320 | 26.315 |
| | CSCS | 14.924 | 14.888 | 21.215 | 21.192 | 27.741 | 27.764 |
| | CFCF | 8.602 | 8.554 | 13.855 | 13.702 | 15.287 | 15.051 |
| | CFSF | 4.465 | 4.412 | 10.199 | 10.172 | 13.450 | 13.347 |
| | CFFF | 0.712 | 0.572 | 3.146 | 3.320 | 5.278 | 5.229 |
| 60 | CCCC | 14.723 | 14.723 | 20.692 | 20.692 | 26.998 | 27.014 |
| | SSSS | 10.126 | 9.842 | 16.709 | 16.687 | 23.483 | 23.464 |
| | CSCS | 11.915 | 11.886 | 18.608 | 18.608 | 25.014 | 25.023 |
| | CFCF | 6.974 | 6.924 | 8.706 | 8.582 | 12.906 | 12.872 |
| | CFSF | 3.786 | 3.741 | 7.619 | 7.585 | 9.982 | 9.961 |
| | CFFF | 0.646 | 0.601 | 2.780 | 2.858 | 4.784 | 4.789 |

**Table 4. The first five dimensionless natural frequencies of an isotropic skew plate with a/b = 1, h/b = 0.2 and β=45 at two different boundary conditions.**

| B.Cs | Ref.s | Modes | | | | |
|---|---|---|---|---|---|---|
| | | 1st | 2nd | 3rd | 4st | 5st |
| CFCF | Present | 2.5708 | 2.6298 | 4.1450 | 5.2714 | 5.8331 |
| | Ref [20] | 2.5712 | 2.6307 | 4.1425 | 5.2744 | 5.8359 |
| | Liew | 2.5674 | 2.6266 | 4.1439 | 5.2627 | 5.8254 |
| CFFF | Present | 0.4206 | 0.9650 | 2.1080 | 2.3900 | 3.6855 |
| | Ref [20] | 0.4212 | 0.9649 | 2.1079 | 2.3903 | 3.6863 |
| | Liew | 0.4218 | 0.9641 | 2.1033 | 2.3866 | 3.6789 |

**Table 5. The dimensionless natural frequencies of SSSS and CCCC orthotropic skew nano plates for first two vibration mode at different nonlocal parameters, aspect ratios b/a and angle β.**

| b/a | β | $(e_0 a)$ (nm) | CCCC | | | | SSSS | | | |
|---|---|---|---|---|---|---|---|---|---|---|
| | | | Ref [23] | | Present | | Ref [23] | | Present | |
| | | | 1st | 2nd | 1st | 2nd | 1st | 2nd | 1st | 2nd |
| 1 | 45 | 0 | 0.0859 | 0.1308 | 0.0868 | 0.1321 | 0.0476 | 0.0889 | 0.0481 | 0.0898 |
| | | 1 | 0.0720 | 0.0949 | 0.0726 | 0.0958 | 0.0408 | 0.0664 | 0.0412 | 0.0671 |
| | 30 | 0 | 0.1301 | 0.1960 | 0.1314 | 0.1980 | 0.0739 | 0.1338 | 0.0746 | 0.1351 |
| | | 1 | 0.0981 | 0.1287 | 0.0991 | 0.1302 | 0.0576 | 0.0908 | 0.0582 | 0.0917 |
| 2 | 45 | 0 | 0.0729 | 0.0812 | 0.0736 | 0.0820 | 0.0372 | 0.0478 | 0.0376 | 0.0483 |
| | | 1 | 0.0645 | 0.0688 | 0.0651 | 0.0695 | 0.0335 | 0.0412 | 0.0338 | 0.0416 |
| | 30 | 0 | 0.1042 | 0.1182 | 0.1052 | 0.1194 | 0.0541 | 0.0718 | 0.0546 | 0.0725 |
| | | 1 | 0.0844 | 0.0916 | 0.0852 | 0.0925 | 0.0450 | 0.0571 | 0.0455 | 0.0577 |

different nonlocal parameters, aspect ratios b/a and corner angle β. the obtained results also have good agreements with reported results in Ref [23].

Influence of nonlocal parameter and damping coefficient of foundation on the variation of fundamental natural frequency of rectangular nanoplate are plotted in Fig 2 and compared with FSDT(present) and CPT (Ref.[9]). According to this figure, The CPT evaluate

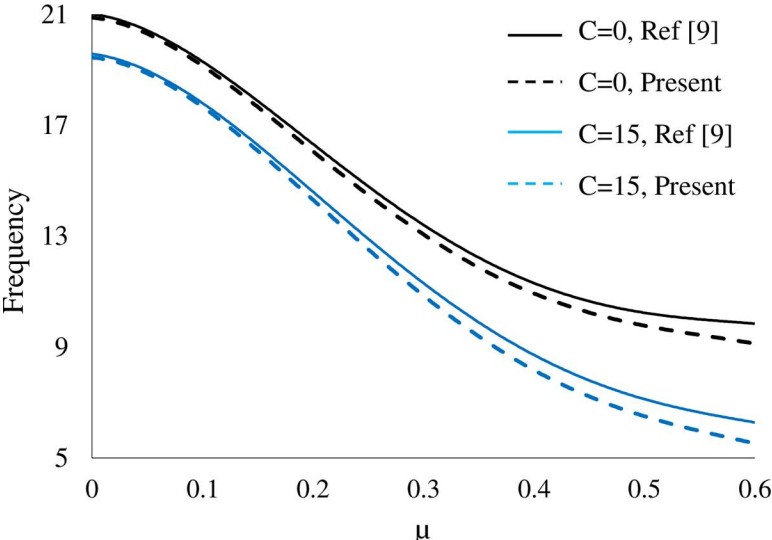

**Fig 2. Fundamental natural frequency of SSSS rectangular nanoplate as a function of nonlocal parameter at different damping coefficient with FSDT(present) and CPT(Ref [ 8]).**

higher values with respect to the FSDT for frequencies and the difference among these values increase with growing the nonlocal parameters and flexibility of nanoplate.

## 4.2 Numerical results

Fig 3 illustrated the variation of fundamental frequencies of trapezoidal nanoplate as a function of bottom angle β, for three different boundary conditions. The geometric and foundation parameters of system are assumed as: a / b = 2 , b = b', K = 100 and C = 5 and another bottom angle is kept

$\gamma = \beta$ . Apparently, fundamental frequencies of both modes decrease when the angle β increases. It can be decided that as the nanoplate has been transformed from oblique (skewed) shape to the rectangular shape (with β=90), the natural frequency decreases, considerably.

Fig 4 investigates the effect of stiffness parameter K on the fundamental frequency of orthotropic CFCF trapezoidal nanoplate at various nonlocal parameter in which the dimensions of the plate are b / a = 0.5 , $\beta = 90, \gamma = 60$ , and foundation damping parameter C = 10. Based on the Fig 4, by increasing the nonlocal parameter and soften the nanoplates, the natural frequencies of nanoplate decreased. Besides, by rising the stiffness coefficient K, the trend of frequencies curves has been increasing. Furthermore, the rate of frequency changes became lower at higher values of K.

Fig 5 illustrates Fundamental frequency variation of CFCF trapezoidal nanoplate as a function of foundation damping parameter (C), when the geometrical parameters of nano plate are: b / a = 0.5 , $\beta = 90, \gamma = 60$ , and foundation stiffness parameter equals to K = 100. By increasing the foundation damping, the frequency curves reduced nonlinearly, which eventually led to zero frequencies and overdamped state. At this condition if the damping parameter increased and became more than a critical damping parameter then the nanoplates did not oscillate. The critical damping parameters are depended on considered nonlocal parameters; for example at μ = 0.1, critical damping equals to $C_{cr}$ = 22.

Effect of the Stiffness and Damping coefficients (K, C) on the fundamental natural frequency of trapezoidal nanoplate is presented in Fig 6 (a-c) at three different boundary conditions (CCCF, CSSS and SFSF, respectively). According to the Figs 6 (a-c), due to the increase of

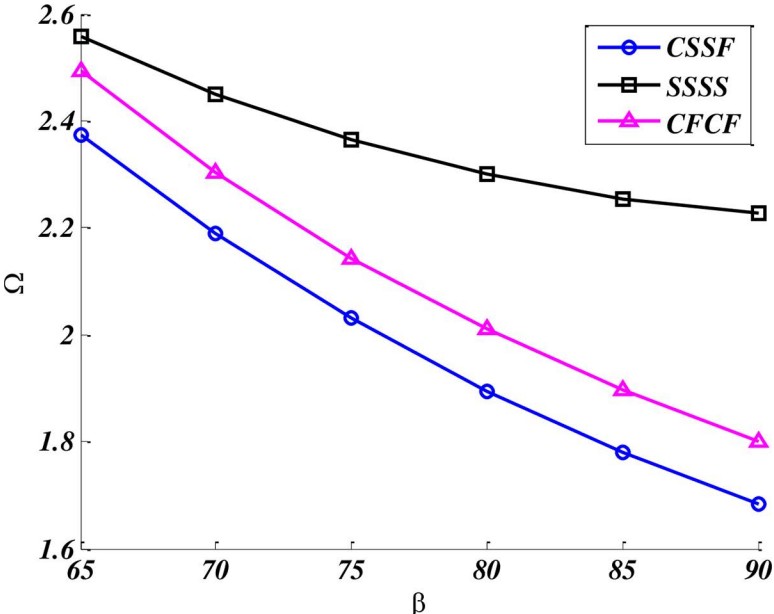

**Fig 3. Variation of fundamental frequency of nanoplate versus angle β = α for different B.Cs.**

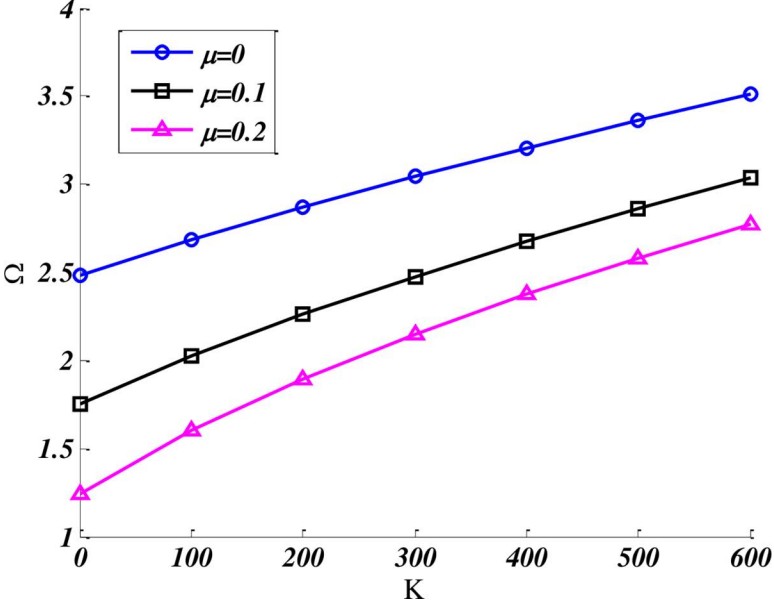

**Fig 4. Effect of stiffness parameter on the fundamental frequency of CFCF quadrilateral nanoplate at various nonlocal parameter.** ( b / a = 0.5 , b' / a = 0.4, $\beta = 60, \gamma = 90$ ), C = 10.

the foundation stiffness, fundamental natural frequency increased, too. Based on the numerical results by increasing foundation's springs parameter (K) the rigidity of the structure, and the natural frequency rises. Actually, resistance against deformation for the nanoplate resting on the foundation became stronger due to increasing of the elastic foundation stiffness. In

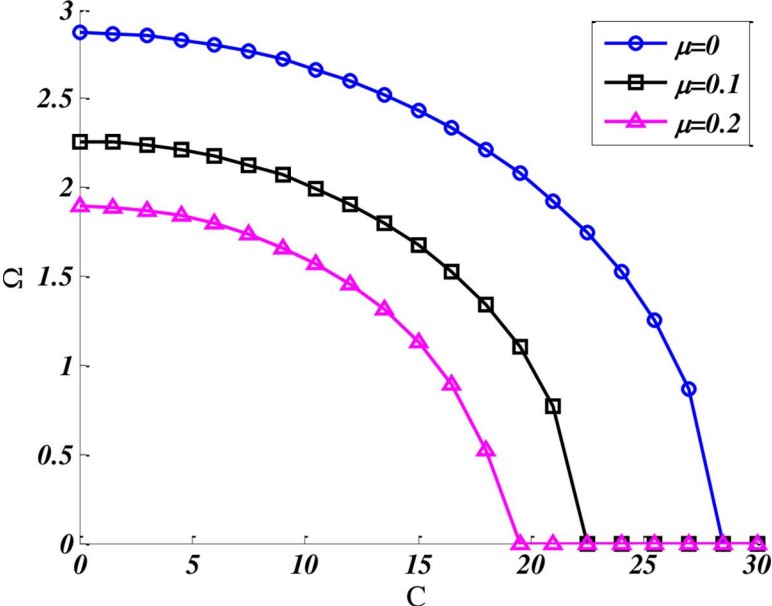

**Fig 5. Effect of damping parameter on fundamental frequency change of both CFCF nanoplate** ( b / a = 0.5 , b' / a = 0.4, $\beta = 60, \gamma = 90$ ), K = 100.

other word, the nanoplate needs to overcome larger resistance to oscillate, so vibrates at a higher frequency.

Besides, as seen at Fig 6c when the damping coefficient became higher, the natural frequency of nanoplate with flexible SFSF boundary conditions reaches to the zero value (overdamped condition), if the stiffness coefficient was smaller than a critical value, these critical values are equals to $K_{cr} = 25$ and $K_{cr} = 200$ for C = 10 and C = 20, respectively.

## 5 Concluding remarks

This article was organized to investigation the vibrational characteristics of trapezoidal nanoplate resting on viscoelastic medium with respect to the influences of different parameters like geometric values of nanoplates, nonlocal parameters and viscoelastic foundation coefficients at different boundary conditions. The governing equations in the Cartesian coordinate system have been transformed into new trapezoidal coordinate; then DQ method have been used for discretizing them at different boundary conditions. The numerical results exposed that, (i) By increasing K, vibration frequencies increased, (ii) numerical results indicate that nonlocal parameter has an important role in the reduction of natural frequencies of vibrational modes; (iii) Increasing the damping parameter of the viscoelastic medium has reduced the natural frequency and this medium has been able to overdamped the oscillations.

The results indicated that adjusting the foundation parameters and BC type of nanoplate can be effective tactics in modifying the vibration behavior of the system. Changing these factors could critically affect vibration features, signifying the opportunity of operating these nano- structures for numerous NEMS applications with specific functionality, such as vibration damping. Also, the new findings can be utilized as a benchmark for upcoming studies about use of quadrilateral nanoplates in real-world engineering scenarios (like nano-sensors, nano-resonators, nano-switches and nanoscale devices working at high frequencies); for example, more particular boundary conditions like elastic BCs with transverse and rotational springs.

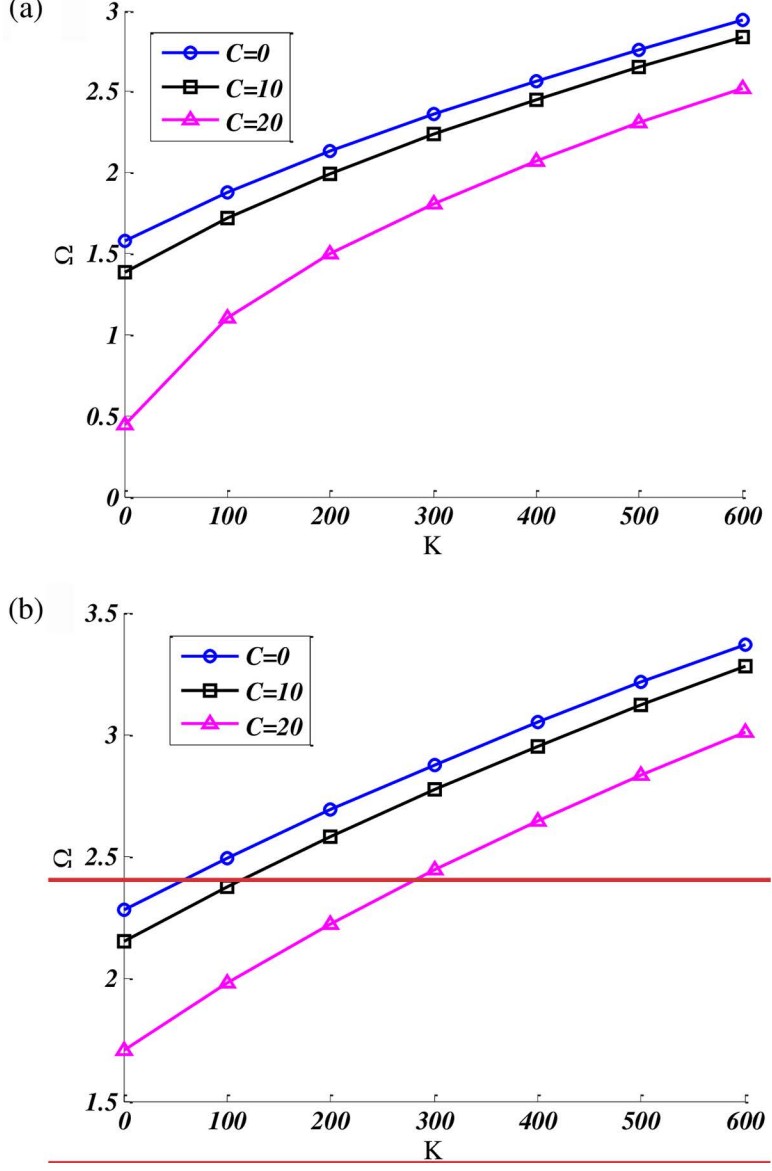

**Fig 6. Variation of fundamental frequencies of nanoplate with respect to C and K, when** $\left(a / b = 1.5, a / b' = 2, \beta = 120, \gamma = 135\right)$ **for different B.Cs;** (a) CCCF, (b) CSSS, (c) SFSF.

While this study developed our understanding of quadrilateral nanoplate vibration and dynamics on viscoelastic foundations, numerous fields of study remain open for more investigation. Current researches (for example: Zhang et al. [37], Cheng et al. [38], Fan et al. [39], Tian et al. [40], Li et al. [41], Song et al. [42], Huang et al. [43] and Wu et al. [44]) demonstrates the serious role of advanced materials, pioneering design organizations, and computational procedures in lecturing current challenges in engineering and vibration analysis.

Also, the investigation of vibration features in viscoelastic systems is mostly related in dynamic platforms, like electro-hydraulic parallel stabilization systems, in which accurate control over vibrations is critical for optimum performance, as reported by Yuan et al. [45]. Moreover, viscoelastic foundations have requests in the railway structures, in which automatic

tracking systems for unfreezing connection wires, as inspected by Du et al. [46], should consider for the dynamic response at changeable circumstances. Also, the vibration behavior lectured in the research is also crucial for estimation of the mechanical characteristics of bolted plates, where damping phenomena changed the load-bearing of system as reported by Cao et al. [47].

## Author contributions

**Conceptualization:** Ramin Abdellahi, Mohsen Esmaeili, Mirsami Yeganli, Roohallah Alizadehsani, Paweł Pławiak.

**Formal analysis:** Ramin Abdellahi, Mohsen Esmaeili, Mirsami Yeganli.

**Funding acquisition:** Paweł Pławiak.

**Investigation:** Mohsen Esmaeili.

**Methodology:** Ramin Abdellahi.

**Project administration:** Roohallah Alizadehsani, Paweł Pławiak.

**Resources:** Ramin Abdellahi, Mirsami Yeganli, Ali Mokhtarian.

**Software:** Mohsen Esmaeili, Mirsami Yeganli.

**Supervision:** Ali Mokhtarian, Paweł Pławiak.

**Validation:** Ramin Abdellahi.

**Visualization:** Ramin Abdellahi.

**Writing – original draft:** Ramin Abdellahi, Mohsen Esmaeili, Mirsami Yeganli.

**Writing – review & editing:** Ali Mokhtarian, Roohallah Alizadehsani, Paweł Pławiak.

## Appendix

The derivatives of any function in the new $\zeta - \eta$ coordinates using the chain rule, are:

$$\begin{Bmatrix} \dfrac{\partial()}{\partial x} \\ \dfrac{\partial()}{\partial y} \end{Bmatrix} = \left[ j \right]^{-1} \begin{Bmatrix} \dfrac{\partial()}{\partial \zeta} \\ \dfrac{\partial()}{\partial \eta} \end{Bmatrix}; \begin{Bmatrix} \dfrac{\partial^2()}{\partial x^2} \\ \dfrac{\partial^2()}{\partial y^2} \\ \dfrac{\partial^2()}{\partial x \partial y} \end{Bmatrix} = \left[ j^{(2)} \right]^{-1} \begin{Bmatrix} \dfrac{\partial^2()}{\partial \zeta^2} \\ \dfrac{\partial^2()}{\partial \eta^2} \\ \dfrac{\partial^2()}{\partial \zeta \partial \eta} \end{Bmatrix} - \left[ j^{(2)} \right]^{-1} \left[ j^{(1)} \right] \left[ j \right]^{-1} \begin{Bmatrix} \dfrac{\partial()}{\partial \zeta} \\ \dfrac{\partial()}{\partial \eta} \end{Bmatrix} \tag{A1}$$

In which () is an arbitrary variable. The components of the transform In which () is an arbitrary variable. The components of the transformation jacobian matrices are:

$$\left[ j \right] = \begin{bmatrix} \dfrac{\partial x}{\partial \zeta} & \dfrac{\partial y}{\partial \zeta} \\ \dfrac{\partial x}{\partial \eta} & \dfrac{\partial y}{\partial \eta} \end{bmatrix}; \left[ j^{(1)} \right] = \begin{bmatrix} \dfrac{\partial^2 x}{\partial \zeta^2} & \dfrac{\partial^2 y}{\partial \zeta^2} \\ \dfrac{\partial^2 x}{\partial \eta^2} & \dfrac{\partial^2 y}{\partial \eta^2} \\ \dfrac{\partial^2 x}{\partial \zeta \partial \eta} & \dfrac{\partial^2 x}{\partial \zeta \partial \eta} \end{bmatrix}; \left[ j^{(2)} \right] = \begin{bmatrix} \left( \dfrac{\partial x}{\partial \zeta} \right)^2 & \left( \dfrac{\partial y}{\partial \zeta} \right)^2 & 2 \dfrac{\partial x}{\partial \zeta} \dfrac{\partial y}{\partial \zeta} \\ \left( \dfrac{\partial x}{\partial \eta} \right)^2 & \left( \dfrac{\partial y}{\partial \eta} \right)^2 & 2 \dfrac{\partial x}{\partial \eta} \dfrac{\partial y}{\partial \eta} \\ \dfrac{\partial x}{\partial \zeta} \dfrac{\partial x}{\partial \eta} & \dfrac{\partial y}{\partial \zeta} \dfrac{\partial y}{\partial \eta} & \dfrac{\partial x}{\partial \zeta} \dfrac{\partial y}{\partial \eta} + \dfrac{\partial x}{\partial \eta} \dfrac{\partial y}{\partial \zeta} \end{bmatrix} \tag{A2}$$

$$\Gamma(\blacksquare) = (\blacksquare) - \mu \left( \begin{array}{c} \left(s_{11}+s_{21}\right)\dfrac{\partial^2(\blacksquare)}{\partial\zeta^2} + \left(s_{12}+s_{22}\right)\dfrac{\partial^2(\blacksquare)}{\partial\eta^2} \\ + \left(s_{13}+s_{23}\right)\dfrac{\partial^2(\blacksquare)}{\partial\zeta\partial\eta} - \left(a_{11}+a_{21}\right)\dfrac{\partial(\blacksquare)}{\partial\zeta} - \left(a_{12}+a_{22}\right)\dfrac{\partial(\blacksquare)}{\partial\eta} \end{array} \right) \tag{A3}$$

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
