## [Decision Letter · Decision Letter 0]

23 Sep 2024

PONE-D-24-34233Free Vibration Characteristics of Trapezoidal Nanoplate Rested on Viscoelastic Substrate With Arbitrary Boundary Conditions Using Differential Quadrature MethodPLOS ONE

Dear Dr. Abdellahi, 

Thank you for submitting your manuscript to PLOS ONE. After careful consideration, we feel that it has merit but does not fully meet PLOS ONE’s publication criteria as it currently stands. Therefore, we invite you to submit a revised version of the manuscript that addresses the points raised during the review process.

Please submit your revised manuscript by Nov 07 2024 11:59PM. If you will need more time than this to complete your revisions, please reply to this message or contact the journal office at plosone@plos.org . Please include the following items when submitting your revised manuscript:

We look forward to receiving your revised manuscript.

Kind regards,

Rab Nawaz

Academic Editor

PLOS ONE

Journal Requirements:

2. In the online submission form you indicate that your data is not available for proprietary reasons and have provided a contact point for accessing this data. Please note that your current contact point is a co-author on this manuscript. According to our Data Policy, the contact point must not be an author on the manuscript and must be an institutional contact, ideally not an individual. Please revise your data statement to a non-author institutional point of contact, such as a data access or ethics committee, and send this to us via return email. Please also include contact information for the third party organization, and please include the full citation of where the data can be found.

4. Please ensure that you refer to Figure 2 in your text as, if accepted, production will need this reference to link the reader to the figure.

**Additional Editor Comments:**

You can find the detailed comments from the experts appended below. Please revise the manuscript in accordance with their feedback.

Reviewers' comments:

Reviewer's Responses to Questions

**Comments to the Author**

1. Is the manuscript technically sound, and do the data support the conclusions?

Reviewer #1: Yes

Reviewer #2: Partly

2. Has the statistical analysis been performed appropriately and rigorously? 

Reviewer #1: N/A

Reviewer #2: Yes

3. Have the authors made all data underlying the findings in their manuscript fully available?

Reviewer #1: Yes

Reviewer #2: No

4. Is the manuscript presented in an intelligible fashion and written in standard English?

Reviewer #1: Yes

Reviewer #2: Yes

5. Review Comments to the Author

Reviewer #1: The paper investigates the free vibration characteristics of trapezoidal nanoplates resting on a viscoelastic substrate using the first-order shear deformation theory (FSDT) and nonlocal elasticity theory, with the differential quadrature (DQ) method as the primary computational tool. This combination of methods addresses an important topic in the field of nanoelectromechanical systems (NEMS), especially in the design and analysis of nanosensors and resonators, as the results can provide significant insights into vibration characteristics at the nanoscale.

The problem is well-framed within the context of modern engineering challenges, such as the need for efficient, sensitive NEMS operating in damped environments. The minor observations are suggested to be incorporated before considering the article for publication.

Reviewer #2: Title

Free Vibration Characteristics of Trapezoidal Nanoplate Rested on Viscoelastic Substrate with Arbitrary Boundary Conditions Using Differential Quadrature Method

Overview

This study investigates the free vibration analysis of trapezoidal nanoplate resting on viscoelastic foundation based on first order shear deformation theory (FSDT) incorporating nonlocal elasticity theory, using differential quadrature (DQ) method.

Comments

• The paper is well-structured but requires further attention to grammar, orthography, and readability. There are some mistakes, sentences to be rewritten and long paragraphs that should be dealt with to improve clarity and comprehension.

• The main contribution of the work is not clearly stated. A clear statement summarizing the main contribution and its relevance to the field should be included in Section 1.

• A paragraph explaining the organization of the manuscript should be added at the end of introduction section. This will help guide readers through the structure of the paper.

• While looking at the broader spectrum of the research, the literature review provided in the manuscript is quite limited particularly in the context of vibration characteristics of structures considered therein. Therefore, I strongly suggest to update the literature review effectively thereby incorporating relevant recent studies to provide a stronger foundation for the research on free vibration characteristics of trapezoidal nanoplate resting on viscoelastic substrates. These recent works contribute into vibration behavior, modeling techniques, and applications, thereby enriching the manuscript.

To enhance understanding of vibration characteristics in complex systems, it is suggested to consider the investigation performed by Yuan et al. (2024), Chinese Journal of Mechanical Engineering, 37(1), 96 https://doi.org/10.1186/s10033-024-01074-w.

To leverage machine learning techniques for vibration analysis, it is vital to incorporate the reference of findings examined by Fan et al (2024) Automation in Construction, 167, 105694 10.1016/j.autcon.2024.105694.

To explore vibration isolation techniques using advanced structures, the work of Tian et al. (2024) Mechanical Systems and Signal Processing, 218, 111587

https://doi.org/10.1016/j.ymssp.2024.111587, on low-frequency vibration isolation using a concave X-shaped structure provides a useful reference for discussing vibration damping mechanisms, which could be particularly relevant when exploring the effects of the viscoelastic substrate in your analysis.

To address advanced force identification methods, Li et al. (2024) Structures, 60, 105840 10.1016/j.istruc.2023.105840, introduced a novel regularization method for identifying moving forces, which could be applied to refine the understanding of external dynamic forces in the nanoplate's vibration behavior.

To consider mechanical properties of structural components, the mechanical behavior of complex structures, as explored by Cao et al. (2024) Construction and Building Materials, 433, 136608 10.1016/j.conbuildmat.2024.136608, provides a useful comparison for understanding how material properties of the nanoplate and substrate interact to influence vibration responses.

To incorporate wave propagation and attenuation models, Wu et al. (2024) Engineering Structures, 315, 118480 https://doi.org/10.1016/j.engstruct.2024.118480 developed an analytical solution for wave propagation in periodic structures, which could provide a theoretical basis for investigating how waves propagate and attenuate in trapezoidal nanoplates.

To incorporate advanced motion control techniques, the motion control strategies explored by Song et al. (2024) IEEE Transactions on Instrumentation and Measurement, 99, 10.1109/TIM.2024.3413202, could provide useful analogies for controlling and understanding vibration dynamics in mechanical structures, particularly in managing dynamic response in nanoplate systems."

To utilize data-driven optimization in structural design, Huang et al. (2022) Engineering Structures, 251, 113479 https://doi.org/10.1016/j.engstruct.2021.113479 demonstrated the use of data-driven optimization for structural design, offering techniques that could be adapted to optimize the boundary conditions and material configurations of trapezoidal nanoplates for improved vibration characteristics.

• Is there any specific reason for not considering more particular boundary conditions like springs, and masses?

• On page 5, change “Mathematical Modeling” to “Mathematical modeling” for consistency. On page 7, ensure that “Mapping and Solution Procedure” is written as “Mapping and solution procedure” to maintain uniformity in the formatting of headings and subheadings. Additionally, correct typographical errors such as “Zhao et al.[17]” to “Zhao et al. [17]” on page 3, and provide only one space after “SLGS [26] to” on page 4.

6. PLOS authors have the option to publish the peer review history of their article (what does this mean? ). If published, this will include your full peer review and any attached files.

**Do you want your identity to be public for this peer review?** For information about this choice, including consent withdrawal, please see our Privacy Policy .

Reviewer #1: No

Reviewer #2: No

---

## [Author Response · Author response to Decision Letter 1]

29 Nov 2024

Manuscript number: PONE-D-24-34233

Title: Free Vibration Characteristics of Trapezoidal Nanoplate Rested on Viscoelastic Substrate with Arbitrary Boundary Conditions Using Differential Quadrature Method

Journal: PLOS ONE

Dear Editor-in-Chief, Dr. Emily Chenette

Thank you for obtaining reviewers’ comments made on our manuscript. We have attempted to modify it accordingly, as highlighted in the revised manuscript. Detailed corrections are listed below point by point:

# Reviewer 1

Reviewer’s comment:

The paper investigates the free vibration characteristics of trapezoidal nanoplates resting on a viscoelastic substrate using the first-order shear deformation theory (FSDT) and nonlocal elasticity theory, with the differential quadrature (DQ) method as the primary computational tool. This combination of methods addresses an important topic in the field of nanoelectromechanical systems (NEMS), especially in the design and analysis of nanosensors and resonators, as the results can provide significant insights into vibration characteristics at the nanoscale.

The problem is well-framed within the context of modern engineering challenges, such as the need for efficient, sensitive NEMS operating in damped environments. The following minor observations are suggested to be incorporated before considering the article for publication:

1. The introduction could benefit from a more detailed explanation of the novelty and main contribution of the research, which is currently lacking. The authors should more clearly outline how their approach is different from previous works and why their focus on trapezoidal nanoplates specifically adds value to the existing body of knowledge.

2. The literature review can be enhanced by incorporating recent studies and replacing outdated references where possible. Specifically, references 17, 18, 19, 35, and 37 can be updated with more current research unless they are essential to cite. To improve the relevance and impact of your references, I suggest replacing a few older studies with the following recent and relevant articles:

• Zhang, J., & Zhang, C. (2023). Using viscoelastic materials to mitigate earthquake-induced pounding between adjacent frames with unequal height considering soil-structure interactions. Soil Dynamics and Earthquake Engineering, 172, 107988. doi: https://doi.org/10.1016/j.soildyn.2023.107988.

• Du, G., Zhang, H., Yu, H., Hou, P., He, J., Cao, S., Ma, L. (2024). Study on Automatic Tracking System of Microwave Deicing Device for Railway Contact Wire. IEEE Transactions on Instrumentation and Measurement, 73, 1-11. doi: 10.1109/TIM.2024.3446638.

• Cheng, Y., Fu, L., Hou, W., Carcione, J. M., Deng, W., Wang, Z. (2024). Thermo-poroelastic AVO modeling of Olkaria geothermal reservoirs. Geoenergy Science and Engineering, 241, 213166. doi: https://doi.org/10.1016/j.geoen.2024.213166.

3. Despite the strengths, the discussion could benefit from a more thorough interpretation of the physical significance of these results. For instance, the role of boundary conditions in practical applications of NEMS is not fully explored. Additionally, the explanation of figures and tables is somewhat lacking in depth; more physical reasoning and detailed analysis would enhance clarity for readers.

4. The conclusions adequately summarize the key findings, particularly the role of stiffness, nonlocal parameters, and damping in influencing natural frequencies. However, the conclusions are rather brief and could be strengthened by offering more in-depth insight into how the findings contribute to advancing the field of nanoscale vibration analysis.

Author's answer:

1. Thank you very much for your attention to the technical part of the study.

To the best of authors’ knowledge, no papers have been reported in the literature concerning the influence of viscoelastic substrate on vibrational behavior of the trapezoidal nanoplate resting on it. In the present study, the size dependent natural frequencies of trapezoidal nanoplate resting on viscoelastic foundation with different combinations of B.Cs have been conducted by means of FSDT and DQ method. The main contribution of the present study is considering the influences of viscoelastic medium in the examination of free vibration behaviour of trapezoidal-shaped nanoplates, having high shape diversity, with different combinations of Boundary conditions (B.Cs).

The reason of our focus on the new finding about trapezoidal shaped nanoplates, are:

Most of the researchers' studies have been investigated the vibration behavior of rectangular and circular nanosheets, and few works have been done on nanosheets with quadrilateral geometry. In 2017, Zhang et al. [1] succeeded in investigating the vibration behavior of nanosheets with a general shape under the effect of an external magnetic field using the Ritz free element numerical method. In this study, the effect of non-local parameter, geometrical angle and boundary conditions on free vibration behavior of parallelogram graphene sheet was investigated. In addition, 12 different states of a single-layer graphene sheet with general geometry were modeled and presented in different boundary conditions (as shown in the next figure).

Figure R1: Schematic of different geometries investigated from the quadrilateral graphene sheet [1]

2.Thank you so much for your kind attention to our study. As mentioned, we remove the older studies (references 17, 18, 19, 35, and 37) and replaced them with never papers, as follows:

• Nguyen Thi et al. [2] (2024) https://doi.org/10.1016/j.heliyon.2024.e26150

• Thom et al.[3] (2024) https://doi.org/10.1016/j.euromechsol.2024.105309

• Torabi et al. [4] (2019) https://doi.org/10.1016/j.euromechsol.2018.07.009

• Wang and Liu [5] (2024) https://doi.org/10.1371/journal.pone.0308245

Moreover, as you noticed the manuscript should be update by replacing new papers in the fields of system’s vibration. The three suggested papers, were in the fields of vibration and published in 2024, added to the manuscript as Refs [37, 38, 46] and highlighted in the revised version.

3. Thank you for your attention to our study. In the case of “interpretation of the physical consequence of the numerical results, such as the role of boundary conditions in practical applications of NEMS”, we add following sentences:

For example: at two different boundary conditions, CFCF (bridge-type) and CFFF (cantilever-type); Based on Table 4, the fundamental natural frequencies of CFCF have been reduced about 84% by removing one clamped BC. Consequently, type of BCs is very crucial in the free vibration behaviour.

Also, we add more physical reasoning and detailed analysis of Table and Figures, as follows:

Figure 6: Based on the numerical results by increasing foundation’s springs parameter (K) the rigidity of the structure, and the natural frequency rises. Actually, resistance against deformation for the nanoplate resting on the foundation became stronger due to increasing of the elastic foundation stiffness. In other word, the nanoplate needs to overcome larger resistance to oscillate, so vibrates at a higher frequency.

Figure 5: By increasing the foundation damping, the frequency curves reduced nonlinearly, which eventually led to zero frequencies and overdamped state. At this condition if the damping parameter increased and became more than a critical damping parameter then the nanoplates did not oscillate. The critical damping parameters are depended on considered nonlocal parameters; for example, at μ=0.1, critical damping equals to Ccr = 22.

Fig. 4: Based on the Fig.4, by increasing the nonlocal parameter and soften the nanoplates, the natural frequencies of nanoplate decreased.

Fig. 3: It can be decided that as the nanoplate has been transformed from oblique (skewed)shape to the rectangular shape (with β=90), the natural frequency decreases, considerably.

Table 4: Based on the results, the fundamental natural frequencies of CFCF have been reduced about 84% by removing one clamped BC. Consequently, type of BCs is very crucial in the free vibration behaviour.

Table 3: The frequency value drops significantly in the plates with replacing clamped boundary with free boundary.

Table 2: By increasing the h/a ratio, the frequency parameter of all modes decreased.

4. Based on your appropriate comment, about “How the findings contribute to advancing the field of nanoscale vibration analysis? “, we complete the conclusion part, as follows:

The results indicated that adjusting the foundation parameters and BC type of nanoplate can be effective tactics in modifying the vibration behavior of the system. Changing these factors could critically affect vibration features, signifying the opportunity of operating these nano- structures for numerous NEMS applications with specific functionality, such as vibration damping. Also, the new findings can be utilized as a benchmark for upcoming studies about use of quadrilateral nanoplates in real-world engineering scenarios (like nano-sensors, nano-resonators, nano-switches and nanoscale devices working at high frequencies); for example, more particular boundary conditions like elastic BCs with transverse and rotational springs.

# Reviewer 2

Reviewer’s comment:

Overview

This study investigates the free vibration analysis of trapezoidal nanoplate resting on viscoelastic foundation based on first order shear deformation theory (FSDT) incorporating nonlocal elasticity theory, using differential quadrature (DQ) method.

Comments

1. The paper is well-structured but requires further attention to grammar, orthography, and readability. There are some mistakes, sentences to be rewritten and long paragraphs that should be dealt with to improve clarity and comprehension.

2. The main contribution of the work is not clearly stated. A clear statement summarizing the main contribution and its relevance to the field should be included in Section 1.

3. A paragraph explaining the organization of the manuscript should be added at the end of introduction section. This will help guide readers through the structure of the paper.

4. While looking at the broader spectrum of the research, the literature review provided in the manuscript is quite limited particularly in the context of vibration characteristics of structures considered therein. Therefore, I strongly suggest to update the literature review effectively thereby incorporating relevant recent studies to provide a stronger foundation for the research on free vibration characteristics of trapezoidal nanoplate resting on viscoelastic substrates. These recent works contribute into vibration behavior, modeling techniques, and applications, thereby enriching the manuscript.

- To enhance understanding of vibration characteristics in complex systems, it is suggested to consider the investigation performed by Yuan et al. (2024), Chinese Journal of Mechanical Engineering, 37(1), 96 https://doi.org/10.1186/s10033-024-01074-w.

- To leverage machine learning techniques for vibration analysis, it is vital to incorporate the reference of findings examined by Fan et al (2024) Automation in Construction, 167, 105694 https://doi.org/10.1016/j.autcon.2024.105694.

- To explore vibration isolation techniques using advanced structures, the work of Tian et al. (2024) Mechanical Systems and Signal Processing, 218, 111587

https://doi.org/10.1016/j.ymssp.2024.111587, on low-frequency vibration isolation using a concave X-shaped structure provides a useful reference for discussing vibration damping mechanisms, which could be particularly relevant when exploring the effects of the viscoelastic substrate in your analysis.

- To address advanced force identification methods, Li et al. (2024) Structures, 60, 105840 https://doi.org/10.1016/j.istruc.2023.105840, offered a novel regularization method for identifying moving forces, which could be applied to refine the understanding of external dynamic forces in the nanoplate's vibration behavior.

- To consider mechanical properties of structural components, the mechanical behavior of complex structures, as explored by Cao et al. (2024) Construction and Build. Materials, 433, 136608 https://doi.org/10.1016/j.conbuildmat.2024.136608, provides a useful comparison for understanding how material properties of the nanoplate and substrate interact to influence vibration responses.

- To incorporate wave propagation and attenuation models, Wu et al. (2024) Eng. Structures, 315, 118480 https://doi.org/10.1016/j.engstruct.2024.118480 developed an analytical solution for wave propagation in periodic structures, which could provide a theoretical basis for investigating how waves propagate and attenuate in trapezoidal nanoplates.

- To incorporate advanced motion control techniques, the motion control strategies explored by Song et al. (2024) IEEE Transactions on Instrumentation and Measurement, 99, https://doi.org/10.1109/TIM.2024.3413202, could provide useful analogies for controlling and understanding vibration dynamics in mechanical structures, particularly in managing dynamic response in nanoplate systems."

- To utilize data-driven optimization in structural design, Huang et al. (2022) Eng. Structures, 251, 113479 https://doi.org/10.1016/j.engstruct.2021.113479 established the use of data-driven optimization for structural design, offering techniques that could be adapted to optimize the boundary conditions and material configurations of trapezoidal nanoplates for improved vibration characteristics.

5. Is there any specific reason for not considering more particular boundary conditions like springs, and masses?

6. On page 5, change “Mathematical Modeling” to “Mathematical modeling” for consistency. On page 7, ensure that “Mapping and Solution Procedure” is written as “Mapping and solution procedure” to maintain uniformity in the formatting of headings and subheadings. Also, correct typographical errors such as “Zhao et al.[17]” to “Zhao et al. [17]” on page 3, and provide only one space after “SLGS [26] to” on page 4.

Author's answer:

1. All of the writing mistakes in the sentences, have been corrected and replaced in the revised manuscript file.

2.The main contribution of the present study is considering the influences of viscoelastic foundation in the examination of free vibration behaviour of trapezoidal-shaped nanoplates, having high shape diversity, with different combinations of Boundary conditions.

3. Thank you for your attention to our study. As you noticed, we add a paragraph explaining the organization of the study at the end of introduction section, as follows:

The present study is organized into four primary sections. Section 2 illustrated the mathematical modeling of nanoplate, the constructive relations and its governing equations of motion. These governing equations have been transformed from the trapezoidal physical domain into the rectangular computational domain in the Section 3 by means of mathematical operations and then rewritten in the form of generalized eigenvalue problem by means of DQ method for various B.C. Finally at section 4, the validation of numerical results presented and parametric study have been shown by means of plots.

4. While thanking your Excellency, as you noticed the manuscript should be update by replacing new papers in the fields of system’s vibration. The eight suggested papers, were in the fields of vibration and published in 2024, added to the manuscript as Refs [39-45, 47] and highlighted in the revised version.

Also, we decided to add another new study in the fields of nanostructure’s vibration for various application as follows:

• Nguyen Thi et al. [2] (2024) https://doi.org/10.1016/j.heliyon.2024.e26150

• Thom et al.[3] (2024) https://doi.org/10.1016/j.euromechsol.2024.105309

• Torabi et al. [4] (2019) https://doi.org/10.1016/j.euromechsol.2018.07.009

• Wang and Liu [5] (2024) https://doi.org/10.1371/journal.pone.0308245

5. Based on your appropriate comment; for the next study we planned to considered general elastic boundary condition for nanoplate with stiffness elements (rotational and transverse springs) in the all directions.

6. Thank you for your attention to our study; all of the mentioned errors have

---

## [Decision Letter · Decision Letter 1]

9 Dec 2024

Free Vibration Characteristics of Trapezoidal Nanoplate Rested on Viscoelastic Substrate With Arbitrary Boundary Conditions Using Differential Quadrature Method

PONE-D-24-34233R1

Dear Dr. Abdellahi,

We’re pleased to inform you that your manuscript has been judged scientifically suitable for publication and will be formally accepted for publication once it meets all outstanding technical requirements.

Kind regards,

Rab Nawaz

Academic Editor

PLOS ONE

Additional Editor Comments (optional):

Reviewers' comments:

Reviewer's Responses to Questions

**Comments to the Author**

1. If the authors have adequately addressed your comments raised in a previous round of review and you feel that this manuscript is now acceptable for publication, you may indicate that here to bypass the “Comments to the Author” section, enter your conflict of interest statement in the “Confidential to Editor” section, and submit your "Accept" recommendation.

Reviewer #1: All comments have been addressed

Reviewer #2: (No Response)

2. Is the manuscript technically sound, and do the data support the conclusions?

Reviewer #1: Yes

Reviewer #2: (No Response)

3. Has the statistical analysis been performed appropriately and rigorously? 

Reviewer #1: N/A

Reviewer #2: (No Response)

4. Have the authors made all data underlying the findings in their manuscript fully available?

Reviewer #1: Yes

Reviewer #2: (No Response)

5. Is the manuscript presented in an intelligible fashion and written in standard English?

Reviewer #1: Yes

Reviewer #2: (No Response)

6. Review Comments to the Author

Reviewer #1: Authors have appropriately addressed the revisions; therefore, I feel no hesitation in suggesting the publication of revised manuscript in Plos one.

Reviewer #2: (No Response)

7. PLOS authors have the option to publish the peer review history of their article (what does this mean? ). If published, this will include your full peer review and any attached files.

**Do you want your identity to be public for this peer review?** For information about this choice, including consent withdrawal, please see our Privacy Policy .

Reviewer #1: No

Reviewer #2: No

---

## [Editor Report · Acceptance letter]

PONE-D-24-34233R1

PLOS ONE

Dear Dr. Abdellahi,

I'm pleased to inform you that your manuscript has been deemed suitable for publication in PLOS ONE. Congratulations! Your manuscript is now being handed over to our production team.

Kind regards,

on behalf of

Dr. Rab Nawaz

Academic Editor

PLOS ONE